# GC-MS and LC-MS Pesticide Analysis of Black Teas Originating from Sri Lanka, Iran, Turkey, and India

**DOI:** 10.3390/toxics11010034

**Published:** 2022-12-30

**Authors:** Kasim Takim, Mehmet Emin Aydemir

**Affiliations:** 1Department of Basic Sciences of Veterinary Medicine, Faculty of Veterinary Medicine, Harran University, Şanlıurfa 63200, Turkey; 2Department of Veterinary Food Hygiene and Technology, Faculty of Veterinary Medicine, Harran University, Şanlıurfa 63200, Turkey

**Keywords:** black tea, pesticide residues, imported tea, Turkey

## Abstract

The purpose of this study is to investigate pesticide residues in the imported and domestic tea sold in Turkey and to detect their compliance with maximum residue limits (MRL) and Acceptable Daily Intake (ADI). A total of 79 samples were analyzed by using LC-MS/MS and GC-MS devices with AOAC 2007.01 method to investigate 603 pesticide residues. According to the results, pesticide residues were found in a total of 28 tea samples. Pesticide residues were found in the countries of origin at the rates as follows: Iran (100%), India (33.3%) and Sri Lanka (17.3%). No pesticide residue was detected in the tea produced in Turkey. The compounds such as Diuron, Ethion, Cypermethrin, Thiacloprid, Thiamethoxam, Fenpyroximate, Acetamiprid, Imidacloprid, Flubendiamide, Deltamethrin and Hexythiazox were detected in positive samples. Seven types of pesticide residues exceeded the MRL determined by the Turkish Food Codex for 15 types (53.57%) for 28 samples with pesticide residue detected. Additionally, 4 types of pesticide residues were determined to exceed the ADI determined by the Codex Alimentarius for 13 (46.42%) of 28 samples with pesticide residue detected. These results have showed that the imported tea entering Turkey was not adequately analyzed in the customs control laboratories or the maximum residual limits were not clearly determined.

## 1. Introduction

Tea is obtained from the leaves of the plant known as *Camellia sinensis.* Tea, which is one of the most important beverages beside water, has a history of 5000 years. Approximately two-thirds of the world’s population consume tea [1,2]. Tea, whose homeland is Assam (India’s interior regions facing towards China), was first grown in China and India [3]. The tea plant is grown in regions with abundant rainfall and hot climate within the zone from about 42 latitudes in the northern hemisphere to 27 latitudes in the southern hemisphere. China, India, Sri Lanka, Kenya, Vietnam, Indonesia, Russia, Japan, Myanmar, Turkey, Bangladesh, Iran, Argentina, Uganda, Tanzania, Malawi, Thailand, Nepal, Rwanda, Burundi, and Ethiopia are among the countries where the tea plant is generally grown [4]. India, China, Sri Lanka, Indonesia, Kenya, and Japan are the countries where the tea plant is commonly grown, and tea production is intense. A total of 80% of tea is produced in these countries [5,6].

Tea growing countries are generally located in the equatorial and near-equatorial regions. The lack of winter season and the humid and hot weather of these regions may lead to infestation, bacterial and fungal diseases in the tea plant. Tea manufacturers use pesticides and herbicides to kill pests, prevent diseases, remove weeds in the tea plantation to supply high quality tea products [7]. Insect infestations occur frequently occurs especially due to the humid climate on the coast of the Caspian Sea where Iran tea is produced. The manufacturers use excessive pesticides to fight the infestation. The people of Iran generally consume broad-leaved pekoe tea produced in Sri Lanka and India. The Iran state exports the tea it produces into its neighboring countries such as Turkey, Azerbaijan, and Russia. Therefore, Iran tea has been reported to seriously threaten the health of the people [8].

The tea plant has benefits such as antioxidant, antimicrobial, anticancer and anti-inflammatory effects on human health, lowering cholesterol and blood pressure and reducing the risk of cardiovascular disease [1,9]. However, pesticide residues in tea products have many harmful effects on health [10]. Pesticide residues by type of the compound, amount and to the shape exposure may lead to serious health problems such as acute effects (irritation, dermatitis, death due to systemic absorption) and chronic effects (lymphoma, leukemia, multiple myeloma, liver cancer, birth defects, neurotoxicity, neurobehavioral disorders, neurophysiologic changes, miscarriage, low birth weight, toxemia, and postpartum bleeding), oxidative stress, mutagenic and carcinogenic effects [11]. Since pesticides pose health risks to consumers and are considered to have potentially harmful effects on the environment, many countries have determined maximum residue levels (MRLs) in tea such as the European Community (EC) Regulation No. 396/2005 [7,12,13].

Various methods are used to determine pesticide residue in foods. However, recently the QuEChERS method (Fast Easy Cheap Effective Robust and Safe) is fast becoming one of the most popular dispersive solid phase extractions (d-SPE) methods in food safety. The QuEChERS approach for ultra-trace identification of pesticides, mycotoxins, and other organic chemicals in food matrices has been widely recognized as accurate and highly efficient [14,15]. Therefore, this method can be used safely in the determination of pesticide compounds in tea samples.

Not all pesticides that contaminate the tea plant may pass into the tea brewing. Actual exposure to pesticides depends on the number of residues that leach into the tea brew. Therefore, when determining the pesticides in the tea plant, the Brewing Factor should be considered to determine the amount that will pass to the consumer [16].

Our aim in this study is to determine the risk assessment for consumer safety with maximum residue limits by determining the suitability of pesticide residues in Iran, Sri Lanka, India, Iraq, Syria, and Turkey teas sold and consumed by the public in Turkey. Scope of work: The answer to the question of "how reliable is the tea from different countries of origin consumed in the field?" will be sought in the sample from Turkey.

## 2. Materials and Methods

### 2.1. Collection and Preparation of Tea Samples for Analysis

Tea samples were obtained by buying from importing companies, from companies that package and wholesale imported teas for distribution, and from tea markets. It was obtained directly from the people living in the villages, especially in the Iran border region. The number of samples to be taken was adjusted according to the population density and consumption frequency of the provinces and the approximate share rates of the countries in the market. The number of samples collected was 79. Iran, Sri Lanka, India, and Turkey were the countries of origin of the samples. In addition, consisted of Sri Lankan origin the black tea that came to Turkey via touristic routes from Germany, Iraq, Syria, and Kuwait. A total of 79 collected samples were distributed by provinces as follows; Mardin; 7 samples, Sirnak; 3 samples, Van; 15 samples, Diyarbakir; 13 samples, Siirt; 9 samples, Batman; 4 samples, Gaziantep; 14 samples, Kilis; 4 samples, Sanliurfa; 10 samples. The number of samples collected was distributed by the countries of origin as follows; Sri Lanka; 52, Iran; 18, Turkey; 6 and India; 3. 

The samples were brought to the laboratory where the analysis was performed. Tea samples were extracted and made ready for LC-MS/MS and GC-MS within the same day. The analyses were carried out by Delta Laboratory Sist. Tezg. San. Tic. Sti. laboratory which is accredited by the European Union. The AOAC 2007.01 method was used for the analysis. The unit is based on mg/kg [8]. Each sample was analyzed in triplicate. Results are given as mean ± standard deviation. Pesticide compounds examined by LC-MS and GC-MS devices are shown in Appendix A.

### 2.2. Extraction of Tea Samples

Extraction of tea samples was performed by modifying the QuEChERS sample preparation procedure [17,18]. After the tea sample was thoroughly homogenized, 2 g was taken and put into a 50 ml centrifuge tube. In order to hydrate the tea, it was vortexed by adding 4 mL of milli-Q water (Millipore, Bedford, MA) and left at room temperature for 30 minutes. Then, 10 mL of acetonitrile was added, and the samples were shaken in an automatic shaker (AGYTAX ^®^, Cirta Lab. SL, Madrid, Spain) for 7 minutes. Then, QuEChERS salt (4 g of magnesium sulphate, 1 g of sodium chloride, 1 g of trisodium citrate dihydrate and 0.5 g of disodium hydrogen citrate sesquihydrate) was added and shaken in a shaker for 7 minutes. The extract was centrifuged at 3700 rpm for 5 min. After this process, 3 mL of the supernatant was taken, transferred to a 15 ml centrifuge tube containing 150 mg of anhydrous calcium chloride and 150 mg of PSA (primary secondary amine), and centrifuged again at 3700 rpm for 5 minutes. A total of 30 μL of 5% formic acid in acetonitrile was added to the extract. From the supernatant, 0.45 μL was taken and filtered, and then given to the device. 

### 2.3. LC-MS Determination of Pesticide Residues

The analysis of pesticide compounds by LC-MS/MS was performed using a Nexera model Shimadzu HPLC attached to a dual MS (triple quadrupole-mass spectrometry) device. The LC-MS/MS system consists of a combination of UHPLC (Shimadzu Nexera model) device and LC-MS (Shimadzu 8040 model triple quadrupole mass spectrometer) device. The liquid chromatography system consists of the LC-30 AD model gradient pump, the DGU-20A3R model degasser, the CTO-10ASvp model column oven and the SIL-30AC model autosampler. Pesticide compounds were tried to be determined in the sample by using the MS Library (SANTE/11312/2021 identification kits were used according to the document) of the device and compared with the compounds registered in the library. The connected equipment and operating conditions are set as follows: Equipment: Two LC-20AD pumps, DGU-20A3R degasser, CTO-10ASVP column oven, SIL-20AC autosampler. Column Information: C-18 Intertsil ODS-4 (3.0 mm x 100 mm, 2 µm) analytical column temperature: 40 °C Mobile phase: A (Water, 0.1% Formic acid), B (Methanol, 0.1% Formic acid). The system delivered a constant solvent flow of 0.3 mL/min and the volume of injection was 2 μL. The elution gradient contains ultrapure water (mobile phase A) and acetonitrile (mobile phase B). To achieve a better chromatographic separation and to facilitate ionization, 10 mM ammonium formate and 0.1% formic acid were added to mobile phase A. The following UHPLC gradient elution program was applied for optimum separation: 5–20% B (0–10 min s), 20% B (10–21 min s), 20–50% B (22–35 min s), 95% B (35–40 min s), 5% B (40–50 min s). 

### 2.4. GC-MS Determination of Pesticide Residues

Some of the pesticides in the tea samples are polar and dissolve in water. Since some of them have non-polar structure, they are included in the class of solutes in non-polar solvents such as hexane and chloroform. Therefore, while LC-MS device is used to analyze polar pesticides, GC-MS device is preferred for nonpolar pesticides. Shimadzu GC-2010, QP-2010 system equipped with triple quadrupole system for separation of pesticide compounds, Shimadzu, Kyoto, Japan DB-FFAP capillary column (60 m × 0.25 mm × 0.25 µm; J&W Scientific, Folsom, CA, USA) was used. The injection temperature of the device was set to 250 °C and the MS ion source to 200 °C. Mass spectrometry was used to scan between 33–650 mass/load (m/z) at 1 second intervals. The temperature program for separation was initiated at 40 °C for 2 minutes and then increased to 240 °C at the rate of 5 mL/min and adjusted for 6 minutes at this temperature. A flow of 1 mL/min was achieved in the column using He as the carrier gas. Wiley 7 and NIST 147 programs in the MS library were used to define the peaks. The retention indices of the defined peaks were determined using C_10_-C_26_ n-alkane series (Dr. Ehrenstorfer GmbH, Ausburg, Germany) and compared with the literature data.

For the optimization process, in order to introduce each analytical standard to the devices, the optimized voltage values that provide the ion transitions and fragmentation conditions according to their molecular weights were determined and MS libraries were created in the devices [19]. 

## 3. Results

Table 1 and Table 2 show the results of the analyses by LC-MS/MS and GC-MS in this study. As seen in Table 1, pesticide residues were found in a total of 28 tea samples collected from the provinces (including Mardin; 4, Van; 15, Sirnak; 1, Siirt; 2, Batman; 1, Diyarbakir; 2, Gaziantep; 2 and Sanliurfa; 1). This means that 36.7% of the collected samples contained pesticide residue. However, this high rate is due to the presence of at least one pesticide in all 18 Iran tea samples (100%). A total of 1 of 3 Indian tea samples (33.3%) and 9 of 52 tea samples in Sri Lanka (17.3%) were found to contain pesticides. Of 603 pesticides studied, only 11 (Diuron, Ethion, Cypermethrin Thiacloprid, Thiamethoxam, Fenpyroximate, Acetamiprid, Imidacloprid, Flubendiamide, Deltamethrin and Hexythiazox) were detected, but the remaining 592 pesticides were not found. 

**Table 1 toxics-11-00034-t001:** LC-MS/MS and GC-MS(/MS) analysis of pesticide quantities determined and risk assessment according to Turkish Codex. Each sample was analyzed in triplicate. Results are given as mean ± standard deviation (mg/kg).

City Where the Tea Sample was Taken	Origin of Tea Sample	Detected Pesticide Compounds	Results/Measurement	Measurement Limit (LOQ)	Uncertainty	Compliance with Regulation	Analysis Method
Mardin	Sri Lanka	Diuron	0.016 ± 0.008	0.010	0.008	Ok	LC-MS/MS
Mardin	Sri Lanka	Diuron	0.021 ± 0.007	0.010	0.014	Not ok	LC-MS/MS
Mardin	Sri Lanka	Diuron	0.011 ± 0.006	0.010	0.005	Ok	LC-MS/MS
Mardin	Sri Lanka	Diuron	0.017 ± 0.006	0.010	0.011	Not ok	LC-MS/MS
Van	Iran	Ethion	0.013 ± 0.007	0.010	0.006	Ok	GC-MS & GC MS/MS
Van	Iran	Cypermethrin	0.014 ± 0.007	0.010	0.007	Ok	GC-MS & GC MS/MS
Van	Iran	Diuron	0.011 ± 0.006	0.010	0.005	Ok	LC-MS/MS
Van	Iran	Ethion	0.017 ± 0.009	0.010	0.008	Ok	GC-MS & GC MS/MS
Van	Iran	Thiacloprid	0.021 ± 0.011	0.010	0.010	Ok	LC-MS/MS
Van	Iran	Thiamethoxam	0.050 ± 0.025	0.010	0.025	Not ok	LC-MS/MS
Van	Iran	Ethion	0.019 ± 0.009	0.010	0.010	Ok	GC-MS & GC MS/MS
		Thiacloprid	0.025 ± 0.013	0.010	0.012	Not ok	LC-MS/MS
		Thiamethoxam	0.054 ± 0.021	0.010	0.033	Not ok	LC-MS/MS
Van	Iran	Thiamethoxam	0.069 ± 0.035	0.010	0.034	Not ok	LC-MS/MS
Van	Iran	Cypermethrin	0.014 ± 0.007	0.010	0.007	Ok	GC-MS & GC MS/MS
Van	Iran	Ethion	0.017 ± 0.009	0.010	0.008	Ok	GC-MS & GC MS/MS
Van	Iran	Fenpyroximate	0.012 ± 0.006	0.010	0.006	Ok	LC-MS/MS
Van	Iran	Thiacloprid	0.036 ± 0.018	0.010	0.018	Not ok	LC-MS/MS
Van	Iran	Acetamiprid	0.062 ± 0.031	0.010	0.031	Not ok	LC-MS/MS
		Cypermethrin	0.027 ± 0.014	0.010	0.013	Not ok	GC-MS & GC MS/MS
Van	Iran	Acetamiprid	0.033 ± 0.017	0.010	0.016	Not ok	LC-MS/MS
		Cypermethrin	0.017 ± 0.009	0.010	0.008	Ok	GC-MS & GC MS/MS
		Ethion	0.012 ± 0.006	0.010	0.006	Ok	GC-MS & GC MS/MS
		Flubendiamide	0.015 ± 0.008	0.010	0.007	Ok	LC-MS/MS
		Thiacloprid	0.080 ± 0.040	0.010	0.040	Not ok	LC-MS/MS
		Thiamethoxam	0.076 ± 0.038	0.010	0.038	Not ok	LC-MS/MS
Van	Iran	Ethion	0.019 ± 0.010	0.010	0.009	Ok	GC-MS & GC MS/MS
		Imidacloprid	0.023 ± 0.012	0.010	0.011	Not ok	LC-MS/MS
		Thiacloprid	0.058 ± 0.029	0.010	0.029	Not ok	LC-MS/MS
		Thiamethoxam	0.076 ± 0.038	0.010	0.038	Not ok	LC-MS/MS
Sirnak	Sri Lanka (Kuwait)	Diuron	0.012 ± 0.006	0.010	0.006	Ok	LC-MS/MS
Siirt	Sri Lanka	Ethion	0.011 ± 0.006	0.010	0.005	Ok	GC-MS & GC MS/MS
		Thiacloprid	0.011 ± 0.006	0.010	0.005	Ok	LC-MS/MS
		Thiamethoxam	0.018 ± 0.009	0.010	0.009	Ok	LC-MS/MS
Siirt	Iran	Diuron	0.028 ± 0.014	0.010	0.014	Not ok	LC-MS/MS
Diyarbakır	Sri Lanka	Deltamethrin	0.029 ± 0.015	0.010	0.014	Not ok	LC-MS/MS
		Thiacloprid	0.047 ± 0.024	0.010	0.023	Not ok	LC-MS/MS
		Thiamethoxam	0.050 ± 0.025	0.010	0.025	Not ok	LC-MS/MS
Diyarbakır	Iran	Hexythiazox	0.015 ± 0.008	0.010	0.007	Ok	LC-MS/MS
		Thiacloprid	0.027 ± 0.014	0.010	0.013	Not ok	LC-MS/MS
		Thiamethoxam	0.052 ± 0.026	0.010	0.026	Not ok	LC-MS/MS
Batman	Iran	Ethion	0.032 ± 0.016	0.010	0.016	Not ok	GC-MS & GC MS/MS
		Thiacloprid	0.086 ± 0.043	0.010	0.043	Not ok	LC-MS/MS
		Thiamethoxam	0.076 ± 0.038	0.010	0.038	Not ok	LC-MS/MS
Gaziantep	India	Ethion	0.021 ± 0.011	0.010	0.010	Ok	GC-MS & GC MS/MS
		Thiacloprid	0.012 ± 0.006	0.010	0.006	Ok	LC-MS/MS
		Thiamethoxam	0.034 ± 0.017	0.010	0.017	Not ok	LC-MS/MS
Gaziantep	Sri Lanka	Ethion	0.020 ± 0.010	0.010	0.010	Ok	GC-MS & GC MS/MS
Sanliurfa	Sri Lanka	Diuron	0.011 ± 0.006	0.010	0.005	Ok	LC-MS/MS

LOQ (mg/Kg); Limit of quantification; : Uncertainty (%); 95% Relative standard uncertainty at confidence level.

**Table 2 toxics-11-00034-t002:** Quantities of pesticides detected by LC-MS/MS and GC-MS analysis and risk assessment according to the ADI values specified by Codex Alimentarius.

City Where the Tea Sample is Taken	Origin of Tea Sample	Detected Pesticide Compounds	Result-Measurement Uncertainty mg/kg	STMR × Dry Tea Consumed Daily (10 mg)	Brew Factor	* TMDI = STMR × Dry Tea Consumed Daily (10 mg) × Brew Factor	ADI (CODEX)
Mardin	Sri Lanka	Diuron	0.008	0.08	-	-	0.01
Mardin	Sri Lanka	Diuron	0.014	0.14	-	-	0.01
Mardin	Sri Lanka	Diuron	0.005	0.05	-	-	0.01
Mardin	Sri Lanka	Diuron	0.011	0.11	-	-	0.01
Van	Iran	Ethion	0.006	0.06	0.025	0.015	0.02
Van	Iran	Cypermethrin	0.007	0.07	0.021	0.015	0.02
Van	Iran	Diuron	0.005	0.05	-	-	0.01
Van	Iran	Ethion	0.008	0.08	0.025	0.002	0.02
Van	Iran	Thiacloprid	0.010	0.10	-		0.01
Van	Iran	Thiamethoxam	0.025	0.25	0.816	**0.566**	0.08
Van	Iran	Ethion	0.010	0.10	0.025	0.0025	0.02
		Thiacloprid	0.012	0.12	0.497	**0.06**	0.01
		Thiamethoxam	0.033	0.33	0.816	**0.27**	0.08
Van	Iran	Thiamethoxam	0.034	0.34	0.816	**0.28**	0.08
Van	Iran	Cypermethrin	0.007	0.07	0.021	0.015	0.02
Van	Iran	Ethion	0.008	0.08	0.025	0.0015	0.02
Van	Iran	Fenpyroximate	0.006	0.06	-	-	0.01
Van	Iran	Thiacloprid	0.018	0.18	0.497	**0.09**	0.01
Van	Iran	Acetamiprid	0.031	0.31	0.806	**0.25**	0.07
		Cypermethrin	0.013	0.13	0.021	**0.28**	0.02
Van	Iran	Acetamiprid	0.016	0.16	0.806	**0.13**	0.07
		Cypermethrin	0.008	0.08	0.021	0.0017	0.02
		Ethion	0.006	0.06	0.025	0.0015	0.02
		Flubendiamide	0.007	0.07	-	-	0.02
		Thiacloprid	0.040	0.4	0.497	**0.2**	0.01
		Thiamethoxam	0.038	0.38	0.816	**0.31**	0.08
Van	Iran	Ethion	0.009	0.09	0.025	0.0023	0.02
		Imidacloprid	0.011	0.11	0.42	0.05	0.06
		Thiacloprid	0.029	0.29	0.497	**0.14**	0.01
		Thiamethoxam	0.038	0.38	0.816	**0.31**	0.08
Sirnak	Sri Lanka (Kuwait) **	Diuron	0.006	0.06	-	-	0.01
Siirt	Sri Lanka	Ethion	0.005	0.05	0.025	0.0013	0.02
		Thiacloprid	0.005	0.05	0.497	**0.025**	0.01
		Thiamethoxam	0.009	0.09	0.816	0.075	0.08
Siirt	Iran	Diuron	0.014	0.14	-	-	0.01
Diyarbakır	Sri Lanka	Deltamethrin	0.014	0.14	0.0046	0.0006	0.01
		Thiacloprid	0.023	0.23	0.497	**0.11**	0.01
		Thiamethoxam	0.025	0.25	0.816	**0.204**	0.08
Diyarbakır	Iran	Hexythiazox	0.007	0.7	-	**-**	0.03
		Thiacloprid	0.013	0.13	0.497	**0.06**	0.01
		Thiamethoxam	0.026	0.26	0.816	**0.2**	0.08
Batman	Iran	Ethion	0.016	0.16	0.025	0.004	0.02
		Thiacloprid	0.043	0.43	0.497	**0.2**	0.01
		Thiamethoxam	0.038	0.38	0.816	**0.3**	0.08
Gaziantep	India	Ethion	0.010	0.10	0.025	0.0025	0.02
		Thiacloprid	0.006	0.06	0.497	**0.03**	0.01
		Thiamethoxam	0.017	0.17	0.816	**0.13**	0.08
Gaziantep	Sri Lanka	Ethion	0.010	0.10	0.025	0.0025	0.02
Sanliurfa	Sri Lanka	Diuron	0.005	0.05	-	-	0.01

TMDI: Theoretical Maximum Daily Intake. STMR: Supervised Trial Median Residues. Brew Factor (BF) = Residues in tea brew ÷ Residues in dry tea leaves. ADI: Acceptable Daily Intake * Calculated according to the data in Table 3. Those exceeding the ADI value are shown in bolt black. ** countries from where tea came to Turkey. —In this country, the Brew Factor for this pesticide has not been reported.

**Table 3 toxics-11-00034-t003:** MRL values of pesticides used in tea according to countries and ADI values according to Codex Alimentarius. Additionally, Transfer rate of pesticide residue from tea brew and Brew Factor.

Pesticides	CODEX MRL (mg/kg)	EU MRL (mg/kg)	USA * mg/kg	Canada * mg/kg	Australia * (mg/kg)	Japan * mg/kg	CODEX ADI mg/kg	Transfer Rate (%)	Water Solubility (mg/L)	Brew Factor **
Diuron	-	0.05	-	-	-	-	0.01	1.4–2.1	0.01	0.014–0.021
Imidacloprid	-	0.05	-	-	-	-	0.06	62.2–63.1	610	0.62–0.63
Fenpyroximate	-	8	-	-	-	-	0.01	-	-	-
Acetamiprid	-	0.05 *	50	-	-	30	0.07	78.3–80.6	4200	0.78–0.81
Cypermethrin	20 (*15)	0.5	-		0.5	20	0.02	-	-	-
Deltamethrin	5	5	-	7	5	10	0.01	0.14–0.46	0.002	0.0014–0.0046
Ethion	-	3	-	-	5	0.3	0.02	2.25–2.5	2	0.022-–0.025
Flubendiamide	50	0.02 *	-	0.02	0.02	40	0.02	-	-	-
Hexythiazox	15	4	-	-	4	35	0.03	-	-	-
Thiamethoxam	20	20	20	-	20	20	0.08	80.5–81.6	4100	0.81–0.82
Thiacloprid	-	10	-	-	10	30	0.01	49.7–50.0	184	0.49–0.50

MRL: Maximum Residue Limits ADI: Acceptable Daily Intake * Indicates lower limit of analytical determination, * Pesticide residue limits in tea by country. ** Brew Factor (BF) = Tea debris residues ÷ Dry tea leaves residues. Transfer of residues into tea brew (%) = BF × 100. —In this country, the MRL for this pesticide has not been reported.

## 4. Discussion

According to the "Turkish Food Codex Regulation on Maximum Residue Limits of Pesticides", the amount of pesticide residue in tea samples should not exceed the maximum residue limit (MRL) of 0.01 mg/kg [12]. According to the results given in Table 1, 15 (53.57%) of 28 samples with pesticide residue detected exceeded this limit. Diuron, Cypermethrin Thiacloprid, Thiamethoxam, Acetamiprid, Imidacloprid and Deltamethrin compounds were determined to be pesticide residues exceeding the MRL. Based on the Codex Alimentarius Commission’s report on Pesticide Residues [13]. Table 2 shows Acceptable Daily Intake (ADI) for some pesticides. According to these results, ADI was exceeded in 13 (46.42%) of 28 samples with pesticide residue detected. Thiacloprid, Acetamiprid and Cypermethrin were seen to be pesticide residues exceeding the ADI values. 

It was determined that the pesticide residue in the tea samples which were produced in Sri Lanka and sold in Mardin was Diuron compound and that 2 of 4 samples with pesticide residue exceeded the maximum residue limits (MRLs). These pesticide residues were seen to be compatible with the Acceptable Daily Intake (ADI) of the Codex Alimentarius for pesticides. It was found that pesticide residues in the imported tea samples which were sold in Van and were all produced in Iran were Diuron, Ethion, Cypermethrin Thiacloprid, Thiamethoxam, Fenpyroximate, Acetamiprid, Imidacloprid and Flubendiamide compounds and that all the samples with pesticide residue exceeded the MRL levels. Moreover, the pesticide residues in eight samples were found to exceed the ADI levels. These pesticides were understood to be Acetamiprid, Cypermethrin, Thiacloprid and Thiamethoxam. Diuron compound was detected in one tea sample from Sri Lanka, packaged in Kuwait, brought into Sirnak province using touristic routes and sold in the stores there, and its amount was found to be at the MRL and ADI level limit. Deltamethrin, Thiacloprid and Thiamethoxam compounds were found to be present in one tea sample produced in Sri Lanka and sold in Siirt, and these residues were detected to be below the MRL and ADI level. Additionally, Diuron compound was observed in one tea sample produced in Iran and sold in Siirt and was found to exceed the MRL level. It was also found that a pesticide compound called Thiacloprid detected in one sample exceeded the ADI level.

Deltamethrin, Thiacloprid and Thiamethoxam compounds were detected in one tea sample produced in Sri Lanka and sold in Diyarbakir and the amount of all compounds were found to be above the MRL level. Additionally, Hexythiazox, Thiacloprid and Thiamethoxam pesticides were found to be present in another tea sample produced in Iran and sold in Diyarbakir, and other two compounds, except Hexythiazox, were determined to exceed the MRL level. In addition, Thiacloprid and Thiamethoxam in two samples were found not to be compatible with the ADI levels.

Ethion, Thiacloprid and Thiamethoxam compounds were detected in one tea sample produced in Iran and sold in Batman, and all the samples were found to be above the MRL level. Further, Thiacloprid and Thiamethoxam in one sample were found not to be compatible with the ADI levels. Ethion, Thiacloprid and Thiamethoxam residues were found in one tea sample produced in India and packaged in Gaziantep; Ethion and Thiamethoxam compounds were understood to exceed the MRL level. Additionally, Ethion compound was found in one tea sample produced in Sri Lanka and packaged in Gaziantep, and it was determined to be at the MRL level limit. In addition, Thiacloprid and Thiamethoxam in two samples were found not to be compatible with the ADI levels. Diuron compound was detected in one tea sample originating from Sri Lanka and purchased in Sanliurfa province. However, the amount of this compound was found to be below the MRL and ADI levels.

We brought the analysis of pesticides in the imported tea consumed in Turkey into [8]. Literature for the first time and no pesticide residue was found in the tea consumed in Sanliurfa in this study. Since the study covered a narrow region, it was required to carry out a new study to include all the cities consuming the imported tea and to analyze more samples. The most interesting result of our study was about the tea produced in Iran. Pesticide residues were found in all the tea produced in Iran and consumed in Turkey and as a result, they were seen to be above the maximum residue limits (MRLs) of determined by the Turkish Food Codex and the Acceptable Daily Intake (ADI) of the Codex Alimentarius Commission. We conducted the literature review and found that the studies on pesticide residues in the tea sold in the stores were too few and inadequate. Therefore, we must admit that the debate to compare our work will be inadequate. However, we consider that this study will guide the literature.

Dehghani et al. [20] conducted the research on pesticides consumed in the Iran market on 2009 and the results showed that "60 types of pesticides were used in Iran. Moreover, they suggested that the responsible authorities should provide the necessary information on the permissible use amounts of pesticides, their protection strategies and the hazards of pesticides on users". The pesticides used in Iran are consistent with the pesticide residues we found. Furthermore, the maximum residue limits were exceeded in all the samples, which means that the pesticide use policy in this country is not developed at the desired level and explains our results. 

According to a report issued by the European Tea Committee on 4 January 1999, it was claimed that the exported Indian tea also contained important levels of pesticide residues [6]. Seenivasan and Muraleedharan [21] conducted large-scale research on the tea produced in tea factories in South India. The researchers investigated some pesticide residues such as dicofol, ethion, quinalphos, hexaconazole, fenpropathrin, fenvalerate and propargite, which were used for pest and disease control in tea in this part of the country in a study conducted with 912 tea samples for three years between 2006 and 2008. The results showed that less than 0.5 percent of tea samples had residues of these pesticides. They also reported that pesticide residues were below their maximum limits in tea as required by the European Union, the Codex Alimentarius Commission of the FAO/WHO and the Government’s Food Safety Modernization Act [19,22]. Our results are parallel to these data for "Ethion and Thiacloprid" in Indian samples, but it is not possible to say the same thing for "Thiamethoxam’’, which the researchers did not analyze. This compound was found to be above the maximum residue limits determined by the Turkish Food Codex. In addition, Thiacloprid and Thiamethoxam compounds in one sample were found to be compatible with the Acceptable Daily Intake (ADI) of Codex Alimentarius for pesticides.

As a result of our analysis, the pesticide compounds in the positive samples were observed to be Ethion, Diuron, Cypermethrin, Thiacloprid, Thiamethoxam, Fenpyroximate, Acetamiprid, Imidacloprid, Flubendiamide, Deltamethrin and Hexythiazox. Many of these compounds were found to seriously threaten human life in literature studies [23,24]. As a result of the analysis, the health effects of pesticides above the maximum residue limit will be discussed in comparison with the literature.

*"Ethion"* compound, a pesticide prohibited for use by the Turkish Food Codex, which was found in 12.65% of the tea samples we collected from the field, was reported to be carcinogenic and lead to especially liver cancer and leads especially to liver hepatitis [25,26]. Although Diuron compound, an herbicide used in agriculture, which was found in 10.12% of the tea samples we analyzed, was slightly toxic to mammals and birds, it was reported that 3,4-dichloroaniline, the main biodegradation product, exhibited a higher toxicity and Diuron compound was toxic to the liver [27,28]. Cypermethrin, which was only found in Iran tea samples taken from Van and was widely used because of its high biological activity and low mammal toxicity, was reported to be a toxic and carcinogenic compound that may lead to various metabolic diseases, especially cancer, by increasing oxidative stress in living things [26,29,30]. Fortunately, this compound was found to be at the maximum residue limit.

Thiamethoxam is a commercially important neonicotinoid insecticide that is absorbed and active against chewing pests. It has been reported that Thiamethoxam is not a mutagen and is not toxic to mammals and does not pose a carcinogenic risk to humans [31,32]. Fenpyroximate is a widely used pesticide. It is known that fenpyroximate inhibits complex 1 in the ATP synthase enzyme complex system, thereby de-energizing living things and having a lethal effect. However, this mechanism leads to the formation of more reactive oxygen species in mammals and therefore, oxidative stress. This oxidative stress was reported to cause many neuro-degenerative diseases including Parkinson disease [31]. It is pleasing that such a harmful pesticide is only present in one sample and below the maximum residue limit. 

Acetamiprid is a strong neonicotinoid insecticide. Animal studies showed that it had low toxicity to mammals. Despite its extensive use, human exposure leading to toxicity is extremely limited [30]. However, a recent study reported that healthy human pesticide evaluated with liver cells (IMR-90) showed both cytotoxic and genotoxic [32]. Unfortunately, this pesticide residue was found in two samples of Iran tea above the maximum residue limit. Imidacloprid, a strong neonicotinoid insecticide, is currently one of the best-selling insecticides. A study, which was aimed to investigate the toxicity of low dose imidacloprid in reproductive organ systems of adult male rats, demonstrated that exposure to imidacloprid led to deterioration of sperm parameters, decreased T level, apoptosis of germ cells, seminal DNA breakdown, depletion of antioxidants and alteration of fatty acid composition, and also suppressed testicular function [33]. This pesticide residue was found in one sample of Iran tea above the maximum residue limit. 

Deltamethrin is a broad-spectrum synthetic pyrethroid insecticide and acaricide commonly used for agricultural and veterinary purposes. However, human and animal exposure have been reported to cause hepatonephrotoxicity [34], leading to nephrotoxic and neurotoxic side effects [35]. This pesticide residue was found in one sample of Sri Lankan tea above the maximum residue limit.

In 2013, the European Food Safety Authority (EFSA) announced the results of analysis of pesticide residues found in food in the European Union (EU). According to the report, 5.1% of the sampled tea, coffee and herbal infusions were above the accepted maximum residue limits [22]. MRL values may vary from country to country. For example, the comparison of the current Codex and consumer country’s MRLs given in Table 3 showed that: The Codex and the United States did not provide any value for "Ethion" used as a pesticide and this value was 3 mg/kg for the European Union, 5 mg/kg for Australia and 0.1 mg/kg for Japan [22]. This limit is accepted to be 0.01 mg/kg for all pesticides in Turkey. So, there is not any specific standard.

According to the Annex 4 of the Turkish Ethion is one of the prohibited pesticides [12]. Although the use of Ethion is prohibited, this compound was found in 10 of 28 samples with pesticide residue according to the results of the analysis. The countries of origin where Ethion is found in the sample are Sri Lanka, India, and Iran.

## 5. Conclusions

There are also Ethion compounds under the prohibited pesticides for use in Turkey specified in Annex 4 of the Turkish Food Codex Regulation on Maximum Residue Limits of Pesticides. Although the use of Ethion is prohibited, this compound was found in 10 of 29 samples with pesticide residue according to the results of the analysis. The countries of origin where Ethion is found in the sample are Sri Lanka, India, and Iran. This result has showed that the imported tea entering Turkey was not adequately analyzed at the customs control points. In addition, Sri Lanka is the world’s largest tea producer and exporter country. Teas produced in Sri Lanka are consumed almost all over the world. The fact that a banned pesticide such as ethion was detected in the teas of this country; moreover, above the MRL limits, is a remarkably interesting result that should be considered. Considering this result, the relevant authorities in Sri Lanka should reconsider the trade of this pesticide. Compared with the samples of other countries, the individual maximum residue limit for each pesticide compound is determined in many countries. Still, it has been seen to be limited to a general limitation of 0.01 mg/kg in our country. Thus, the pesticide MRL in tea does not have a general limit of 0.01 mg/kg for all pesticides but needs to be individually determined for each compound. In addition, these results showed that teas originating in Iran contain serious amounts of pesticides. Therefore, its consumption can be dangerous for health. We consider this situation to be an important warning for both Iranian people and other people who consume Iranian tea.

## Data Availability

Not applicable.

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
