# Peer review of "GC-MS and LC-MS Pesticide Analysis of Black Teas Originating from Sri Lanka, Iran, Turkey, and India"

_toxics, 2022, doi:10.3390/toxics11010034_

Round 1

Reviewer 1 Report (New Reviewer)

Line 43: Please clarify "Heavy infestation". 

Line 52-58: This sentence is too generic. Althoough agree with the negatice effects of pesticides, those are strongly corellated with the kind of each compound as regards its toxicity and other characteristics, the application pattern followed in agricultural practice, the polupation group (adults, childre, etc) as well as food consumption data.  

The scope of the present study should be cleared stated. To my understanding the estimation of compliance with the in force MRLs of pesticide residues monitored in tea samples is not enough for a scientific paper. 

However, monitoring of pesticides and risk assessment for consumer safety is a very important and crucial field anf therefore, authors should recondider and inprove their manuscript. 

Author Response

Manuscript ID: toxics- 2089039

Title     : Pesticide Analysis in Sri Lanka, Iran, Turkey and India Origin
Black Teas by LC-MS and GC-MS

Responses to Reviewer

First, I want to thank you for your informative and necessary comments for my manuscript. I have tried to do necessary modifications in light of your recommendations accordingly and detailed corrections are listed below point by point. Thank you for your interest.

Reviewer #1:

Q1. Line 43: Please clarify "Heavy infestation". 

R1. The necessary corrections were done in the text according to the reviewer’s suggestion.

Q2. Line 52-58: This sentence is too generic. Althoough agree with the negatice effects of pesticides, those are strongly corellated with the kind of each compound as regards its toxicity and other characteristics, the application pattern followed in agricultural practice, the polupation group (adults, childre, etc) as well as food consumption data.  

R2. The necessary corrections were done in the text according to the reviewer’s suggestion. Changes are shown in blue.

Q3. The scope of the present study should be cleared stated. To my understanding the estimation of compliance with the in force MRLs of pesticide residues monitored in tea samples is not enough for a scientific paper. However, monitoring of pesticides and risk assessment for consumer safety is a very important and crucial field anf therefore, authors should recondider and inprove their manuscript. 

R3. The necessary corrections were done in the text according to the reviewer’s suggestion. The article has been thoroughly revised. Changes made are shown in blue in the text.

Reviewer 2 Report (New Reviewer)

The present manuscript entitled "Pesticide Analysis in Sri Lanka, Iran, Turkey and India Origin Black Teas by LC-MS and GC-MS" by Kasım Takım and Mehmet Emin Aydemir (toxics-2089039) is written correctly and has a good structure; moreover, it has all the necessary parts. The article is interesting from an environmental point of view; therefore, it should interest the reader. I proposed improvements in the method description and with a presentation of results. The paper meets Toxics' requirements, and I recommend the article for publication in Toxics following the common editing stage. My current decision is a major revision. More specific comments and observations are presented below.

1. Introduction. It seems a good solution to add a paragraph on analytical methods used to test pesticide residues.

2. Why are some parts highlighted in yellow?

3. Tables in SI should be named as Table S1 and S2. There are a lot of typos in these tables, and they need to be corrected.

4. Page 2, line 93. What were the parameters of the water used?

5. Please check the record of units and unify them (“l” or “L”).

6. Please check that all abbreviations are derived before using them.

7. Sometimes strange letters appear in the text.

8. Section 2.3. Was a gradient used? What did such a gradient look like during the analysis?

9. The MS measurement parameters should be described in more detail.

10. Table 1. What is the Measurement Limit? Is it LOD? How was it calculated? The next column shows Uncertainty. What is that uncertainty? How was it determined?

11. Please correct the typos in the text.

12. Please check whether the use of bold in the text and tables complies with the journal's requirements.

13. Exemplary chromatographs can be added. It will be interesting for the reader.

14. Page 8, lines 285, 290, and page 9, lines 301, 302. Commas in values must be converted to dots.

15. Does the conducted studies have disadvantages?

16. Conclusions. This part definitely needs expanding.

17. References. This part definitely needs improving. Sometimes there is a doi number, and sometimes not. Journals are written sometimes as a full name and sometimes as an abbreviation.

I hope that the comments presented will help improve the article.

Author Response

Manuscript ID: toxics- 2089039

Title     : Pesticide Analysis in Sri Lanka, Iran, Turkey and India Origin
Black Teas by LC-MS and GC-MS

Responses to Reviewer

First, I want to thank you for your informative and necessary comments for my manuscript. I have tried to do necessary modifications in light of your recommendations accordingly and detailed corrections are listed below point by point. Thank you for your interest.

Reviewer #2:

Q1. Introduction. It seems a good solution to add a paragraph on analytical methods used to test pesticide residues.

R1. The corrections were done in the text according to the reviewer’s suggestion. All the changes requested by the referee were made.

Q2. Why are some parts highlighted in yellow?

 R2. Correction has been requested before. It was made to indicate the corrected places.

Q3.  Tables in SI should be named as Table S1 and S2. There are a lot of typos in these tables, and they need to be corrected.

R3. The molecule names in the table are mostly new and give a typo because word cannot define them. We think there are no misspellings.

Q4. Page 2, line 93. What were the parameters of the water used?

R4. corrected in text

Q5. Please check the record of units and unify them (“l” or “L”).

R5. The corrections were done in the text according to the reviewer’s suggestion.

Q6.  Please check that all abbreviations are derived before using them.

R6. The corrections were done in the text according to the reviewer’s suggestion.

Q7. Sometimes strange letters appear in the text.

R7. -corrected in text

Q8. Section 2.3. Was a gradient used? What did such a gradient look like during the analysis? The MS measurement parameters should be described in more detail.

R8. Yes, it was used and the relevant method was added to the text.

      Explained. Changes were shown in blue in the text.

Q9.  A) Table 1. What is the Measurement Limit? Is it LOD? How was it calculated?

  1. B) The next column shows Uncertainty. What is that uncertainty? How was it determined?

R9.  A) The limit of detection we use is the LOQ. The term LoQ stands for quantization limit. It gives the smallest concentration of an analyte in a test sample that we can determine with acceptable repeatability and accuracy. In other words, it is the concentration at which the entire analytical system should yield a recognizable signal and an acceptable calibration point. LOQ= 10 x SD/m equation is used.

  1. B) Uncertainty (%); 95% Relative standard uncertainty at confidence level,. Uncertainty of measurement is an indicator of confidence in analytical data and refers to the range around a reported or experimentally obtained result within which the true value is expected to fall within a certain probability (confidence level).

What the LOQ and U values represent is given below the Table1.

Q10.  Please correct the typos in the text.

R10. -corrected in text

Q12. Please check whether the use of bold in the text and tables complies with the journal's requirements. Please check whether the use of bold in the text and tables complies with the journal's requirements.

R12. -corrected in text

Q13. Exemplary chromatographs can be added. It will be interesting for the reader.

R13.  Chromatograms of the pesticide samples can be added to the article. However, we think that this is not necessary. Because pesticides were detected in 29 samples. It is necessary to obtain a separate chromatogram for each device. The inclusion of these in the article will seriously increase the volume of the article. Instead, we will present pesticide reports as evidence in the supplementary materials section.

Q14. Page 8, lines 285, 290, and page 9, lines 301, 302. Commas in values must be converted to dots.

R14. -corrected in text

  1. 15. Does the conducted studies have disadvantages?

R15. There is no downside to the work done. On the contrary, it has produced very advantageous results in terms of informing the consumer. For example, teas of Iranian origin should be consumed more cautiously, etc.

Q16. Conclusions. This part definitely needs expanding.

R16. Conclusions expanded

Q17. References. This part definitely needs improving. Sometimes there is a doi number, and sometimes not. Journals are written sometimes as a full name and sometimes as an abbreviation.

R17. The corrections were done in the text according to the reviewer’s suggestion.

Reviewer 3 Report (New Reviewer)

General comments

The paper is well structured and the reader is able to follow aim, results and discussion of the investigations. Nevertheless, there are some questions to be asked and a number of corrections mainly concerning the English language which are addressed under the paragraph “Special comments”.

Samples were “collected”. Does this mean, someone went to the market and bought the samples? Or were they delivered directly by the producing or distributing country? Or were they collected at the customs? Please specify.

In paragraph 2.4, a triple quadrupole GC-MS system is mentioned; in Table 1, GC-MS and GC-MS/MS are listed in parallel. Nowhere is described which one was used in which case(s). Or was the same instrument used in different modes? The text says that a scan mode was involved which would be GC-MS. The table presents concentration results which should be better determined by GC-MS/MS. So, please specify and provide more details of the GC-MS(/MS) methods.

In paragraph 2.3 the MS system of the LC-MS/MS is missing. Only the HPLC instrument is specified as a Shimadzu instrument. Please provide information about the mass spectrometer used in this study incl. MS parameters as i.e. ionization mode. Further missing information is about the gradient applied for LC separation. Please provide details of the gradient or whether an isocratic method was applied.

There were more than 600 pesticides involved in this study (fortunately, only 11 were really found in the tea samples). In case of volatile and less polar compounds I assume that they were screened with GC-MS. But what about those pesticides which are highly polar and less volatile? “Normal” LC-MS/MS (i.e. using a triple quadrupole instrument) is targeted analysis which is not suitable for unknown screening. How did you do this? Did you use an Orbitrap instrument for screening? Or did you indeed perform targeted analyses meaning that the instrument was fed with MRM transition data of hundreds of pesticides? This is not clearly communicated in the manuscript. Please provide additional information.

Table 1, column 7: this column is needless if the unit is provided with the concentration results (column 4); the headline should then be: c [mg/kg].

Table1, column 8: The word “suitable” seems not to be the best expression in this context. I would suggest to simply use “ok” or “not ok”.

Paragraph 2.2 describes the chemical process of pesticide extraction from the tea leaves. But what about the brewing process? Was it exactly the same as for preparing tea as a hot beverage? Please provide some short information about this as you give a number of details concerning solubility in water (hot water or cold water?) and the brewing factor. May be that some pesticides do not survive brewing with hot water. This fact would also be interesting for the reader to know.

Special comments

Title: I would suggest a little change: GC-MS and LC-MS Pesticide Analysis of Black Teas Originating from Sri Lanka, Iran, Turkey, and India

Page 1, line 28: …beverages beside water

Page 1, line 29: …population consume tea

Page 1, line 31: remove “in the world”

Page 1, line 32: within the zone…

Page 1, lines 42 and 43: …weeds in the tea plantation to provide high…

Page 2, lines 75 – 79: This sentence is unclear. Please reword.

Page 3, line98: 3 mL

Page 3, line 100: What is PSA? Please explain this abbreviation.

Page 3, line 101: 0.45 µm cannot be the correct unit; was it µL or mL?

Page 3, line108: device and compared

Page 3, line 111: remove Column

Page 3, line 112: use small letter for gradient composition

Page 3, line 113: end line with a full stop

Page 3, line 121: 5 °C

Page 3, line134, last word: contained

Page 3, line 135: pesticide residues; remove hyphen between high and rate (otherwise the reader sees the sentence as incomplete).

Page 3, lines 135 and 136: connect both sentences: …from Iranian tea because at least …

Page 3, line 137: …to contain pesticides

Page 3, line 138: …pesticides studied, only 11…

Page 3, line 140 - suggestion: detected instead of observed

Page 3, headline Table 1: … and GC-MS / MS or GC-MS(/MS) (see above)

Page 4, Table1, headline of column 1: tea sample was taken

Page 4, Table 2, headlines of columns 5 and 7: space between daily and (10 mg)

Page 5, line 148: ...countries from where….

Page 6, lines 161 – 162: The sentence after MRL is incomplete. Please reword or at least add a suitable verb.

Page 6, line176 – 177: spelling of Kuwait is different in Table 2 – line Sirnak / Sri Lanka; please unify

Page 7, line 202 – 204: The sentences are incomplete and cannot be understood; please reword.

Page 7, line 214: remove space after research

Page 8, line 246: … and leads especially to liver… hepatitis

Page 8, lines 256 – 257: absorbed

Page 8, lines 268 – 269: This sentence seems to be wrong: Not the liver cells are cytotoxic but the pesticide tested with liver cells showed cytotoxic effects, isn’t it? Please reword accordingly.

Page 8, lines 292 – 294: Suggestion for rewording of this sentence: According to the Annex 4 of the Turkish… Ethion is one of the prohibited pesticides.

Page 9, lines 305 – 306, last word: Do you mean laboratory?

Page 10, line 355 (Ref. 18): Volume is 2. This was found by accident, so please check whether volume numbers and pages are given correctly.

Author Response

Manuscript ID: toxics- 2089039

Title     : Pesticide Analysis in Sri Lanka, Iran, Turkey and India Origin
Black Teas by LC-MS and GC-MS

Responses to Reviewer

First, I want to thank you for your informative and necessary comments for my manuscript. I have tried to do necessary modifications in light of your recommendations accordingly and detailed corrections are listed below point by point. Thank you for your interest.

Reviewer #3:

Q1. Samples were “collected”. Does this mean, someone went to the market and bought the samples? Or were they delivered directly by the producing or distributing country? Or were they collected at the customs? Please specify.

R1. The corrections were done in the text according to the reviewer’s suggestion. Changes were shown in blue in the text.

Q2. In paragraph 2.4, a triple quadrupole GC-MS system is mentioned; in Table 1, GC-MS and GC-MS/MS are listed in parallel. Nowhere is described which one was used in which case(s). Or was the same instrument used in different modes? The text says that a scan mode was involved which would be GC-MS. The table presents concentration results which should be better determined by GC-MS/MS. So, please specify and provide more details of the GC-MS(/MS) methods.

 R2. Some of the pesticides in the tea samples are polar and dissolve in water; Since some of them have non-polar structure, they are included in the class of solutes in non-polar solvents such as hexane and chloroform. Therefore, while LC-MS device is used to analyze polar pesticides, GC-MS device is preferred for nonpolar pesticides. The corrections were done in the text according to the reviewer’s suggestion. Changes were shown in blue in the text.

Q3.  In paragraph 2.3 the MS system of the LC-MS/MS is missing. Only the HPLC instrument is specified as a Shimadzu instrument. Please provide information about the mass spectrometer used in this study incl. MS parameters as i.e. ionization mode. Further missing information is about the gradient applied for LC separation. Please provide details of the gradient or whether an isocratic method was applied.

R3. Detailed information suggested by the referee is given in the text. Changes are indicated by color difference.

Q4. There were more than 600 pesticides involved in this study (fortunately, only 11 were really found in the tea samples). In case of volatile and less polar compounds I assume that they were screened with GC-MS. But what about those pesticides which are highly polar and less volatile? “Normal” LC-MS/MS (i.e. using a triple quadrupole instrument) is targeted analysis which is not suitable for unknown screening. How did you do this? Did you use an Orbitrap instrument for screening? Or did you indeed perform targeted analyses meaning that the instrument was fed with MRM transition data of hundreds of pesticides? This is not clearly communicated in the manuscript. Please provide additional information.

R4. The analysis method was created by introducing each pesticide sample to the device one by one. The laboratory we work with is a serious institution that has received European Union accreditation. Orbitrap-based MS system is not used. The method and the detectors used are very sensitive anyway. Recovery and LOQ values are given for each pesticide. We recommend that you review the Report we have presented in the supplementary materials.

Q5. Table 1, column 7: this column is needless if the unit is provided with the concentration results (column 4); the headline should then be: c [mg/kg].

R5. The corrections were done in the text according to the reviewer’s suggestion.

Q6. Table1, column 8: The word “suitable” seems not to be the best expression in this context. I would suggest to simply use “ok” or “not ok”.  .

R6. - corrected in text

Q7. Paragraph 2.2 describes the chemical process of pesticide extraction from the tea leaves. But what about the brewing process? Was it exactly the same as for preparing tea as a hot beverage? Please provide some short information about this as you give a number of details concerning solubility in water (hot water or cold water?) and the brewing factor. May be that some pesticides do not survive brewing with hot water. This fact would also be interesting for the reader to know

R7.  - In this study, the brewing process was not carried out in the tea samples. Therefore, it is considered inappropriate to make such a statement. The corrections were done in the text according to the reviewer’s suggestion.

Round 2

Reviewer 1 Report (New Reviewer)

Line 130, 131 and 133: Please correct with  movement phase with "mobile phase". Additional the term is chromatographic separation and not differentiation. 

Lines 137-139: Please rephrase "Some of the pesticides in the tea samples are polar and dissolve in water; Since some 137 of them have non-polar structure, they are included in the class of solutes in non-polar 138 solvents such as hexane and chloroform." as to be more comperhensive. 

Conclusions should be reorganized. 

Author Response

Thank you for your very valuable contribution. I also made the final corrections you gave and showed them in blue in the text.

Reviewer 2 Report (New Reviewer)

Dear Authors,

Thank you for your meticulous consideration of my comments. The paper has improved substantially and, in my opinion, is suitable for publication.

Author Response

You have made very useful contributions to the improvement of our article. I would also like to thank you for your valuable contributions and wish you healthy and peaceful days.

This manuscript is a resubmission of an earlier submission. The following is a list of the peer review reports and author responses from that submission.

Round 1

Reviewer 1 Report

The aim of the study was to determine residues of 603 pesticides in 79 samples of black tea. The use of pesticides is necessary to ensure adequate protection of crops from insects, for example, however the chemical residues in food are significant source of human exposure. Therefore, it is important to monitor the concentration of pesticide residues in food to assess the risk to human health. The authors had the right goal, but conducting a quantitative analysis requires validation of the method using analytical standards. My main concern is, how did the authors conduct a risk assessment without a properly performed quantification?

The description of the methodology lacks a lot of key information and is not entirely clear - the description of sample preparation should include more details, did the authors confirm the library-identified pesticides with analytical standards, how was the method validated (precision, accuracy, recovery, etc.), how were calibration curves prepared, how was quality control performed?

The manuscript contains many errors and requires major revision to be published. Below are some detailed comments:

Section "2. Materials and Methods" - please correct the numbering of the subsections

Line 82, section "2.1. Determination of pesticide residues by LC-MS," section "3.2. Determination of pesticide residues by GC-MS": in line 82, the authors wrote that they used LC-MS and GC-MS, while LC-MS/MS and GC-MS/MS are also listed in Table 1. Which technique did the authors use? If MS/MS, did the authors optimize the fragmentation conditions and how were the pesticides identified, as the identification was based on MS libraries and not on analytical standards? What mass spectrometer was used, was it a triple quadrupole? The description of the detection technique needs more details.

Line 86: "a dual MS device" - please add more details about the mass spectrometer

Line 87: What type of MS library did the authors use? What pesticide identification criteria were used?  

Line 91: "Column temperature: 400 C" - Do the authors mean 40°C? Please correct

Line 96: What kind of salt was used?

Line 98, Line 114: "a 0.45 L filter" please provide the correct filter pore size

Line 104: "mass/load (m/z)" please change to mass-to-charge ratio

Line 105: "the rate of 5 mL/min" - Do the authors mean 5°C/min? Please correct

Line 107-108: What pesticide identification criteria were used?

Line 112: What does "the clean-up kit tube" contain?

Table 1: What do the authors mean by "measurement limit" and "uncertainty"? What are the results in the "Results/Measurement" column, is it the mean and error value? If mean, what kind of measurements are involved, several repetitions for one sample? What kind of error is it, the standard deviation?

Please edit the table to clearly show which row shows the results for the next sample.

Table 2: What is a brew factor? Please provide a definition in the text as well as references for these values

Table 3: please unify the units notation. What do the names of the countries mean, are they reference values or concentrations in tea samples obtained in studies from these countries? Please add references for the values
